# Nanovaccines for Cancer Prevention and Immunotherapy: An Update Review

**DOI:** 10.3390/cancers14163842

**Published:** 2022-08-09

**Authors:** Xingliang Fang, Huanrong Lan, Ketao Jin, Daojun Gong, Jun Qian

**Affiliations:** 1Department of Hepatobiliary Surgery, Affiliated Hospital of Shaoxing University, Shaoxing 312000, China; 2Department of Breast and Thyroid Surgery, Affiliated Jinhua Hosptial, Zhejiang University School of Medicine, Jinhua 321000, China; 3Department of Colorectal Surgery, Affiliated Jinhua Hosptial, Zhejiang University School of Medicine, Jinhua 321000, China; 4Department of Gastrointestinal Surgery, Affiliated Jinhua Hospital, Zhejiang University School of Medicine, Jinhua 321000, China; 5Department of Colorectal Surgery, Xinchang People’s Hospital, Affiliated Xinchang Hosptial, Wenzhou Medical University, Xinchang 312500, China

**Keywords:** nanovaccines, cancer therapy, immunotherapy

## Abstract

**Simple Summary:**

Cancer vaccines are a promising immunotherapy-based agents used in cancer therapy. However, monotherapy with these vaccines does not have the sufficient effectiveness in clinical settings. To overcome this challenge, researchers designed nanosystems that increase cancer vaccine efficacy and effectiveness by improving the vaccine's half-life and durability, inducing TME reprogram-ming, and enhancing the anti-tumor immunity with minimum toxicity. This review summarized the structure and different types of cancer nanovaccines and their mechanisms of action in cancer therapy. Moreover, the advantages and drawbacks of these vaccines are discussed.

**Abstract:**

Cancer immunotherapy has received more and more attention from cancer researchers over the past few decades. Various methods such as cell therapy, immune checkpoint blockers, and cancer vaccines alone or in combination therapies have achieved relatively satisfactory results in cancer therapy. Among these immunotherapy-based methods, cancer vaccines alone have not yet had the necessary efficacy in the clinic. Therefore, nanomaterials have increased the efficacy and ef-fectiveness of cancer vaccines by increasing their half-life and durability, promoting tumor mi-croenvironment (TME) reprogramming, and enhancing their anti-tumor immunity with minimal toxicity. In this review, according to the latest studies, the structure and different types of nanovaccines, the mechanisms of these vaccines in cancer treatment, as well as the advantages and disadvantages of these nanovaccines are discussed.

## 1. Introduction

In recent decades, many approaches based on immune system manipulation have been introduced to treat cancer, and some of these methods have had successful results in their clinical phases [1]. Cancer treatment using cancer vaccines is one of these novel methods that has continuously been considered because the purpose of this type of vaccine is to create strong anti-tumor immune responses, the specific elimination of tumor cells with minimal damage to non-tumor cells, along with the creation of immunological memory against tumor antigens [2]. However, the outcomes of clinical studies show that these types of vaccines have not yet been able to find a suitable place in cancer immunotherapy due to the development of weak and short-lived anti-tumor immune responses [3].

Studies are ongoing to find solutions to the challenges of cancer therapy using cancer vaccines, and it has been shown that formulating vaccines with delivery vehicles such as nanoparticles (NPs) can enhance antigen delivery to antigen-presenting cells (APCs) and improve antigen presentation by APCs to effector lymphocytes [4]. Nowadays, nanovaccines are widely used to treat cancer and infectious diseases such as COVID-19 [5,6]. The main goals of nanovaccines in cancer treatment are to activate the anti-tumor immune responses and inhibit the immunosuppressive responses in the tumor microenvironment (TME) [7,8]. These vaccines increase antigen stability by encapsulating antigens in nanocarriers and prevent their degradation. Moreover, the use of adjuvants along with antigens can facilitate the co-delivery and capture process by APCs, thereby improving the immunogenicity and stability of the vaccine. Some NPs are designed for the cytoplasmic delivery of antigens, which can be used to enhance anti-tumor responses of CD8^+^ cytotoxic T cells through cross-presentation of the antigen by major histocompatibility complex I (MHC-I). Surface modifications of NPs with the aim of specific immunomodulation and using polyvalent antigens at the nanoparticle surface can help induce humoral and B cell-dependent responses [9].

Recently, it was reported that biomimetic cytomembrane nanovaccines could have long-term anti-tumor immunity and decrease regulatory T cells (Tregs), increasing the frequency of CD8^+^ T cells in the tumor and increasing spleen effector memory T cells. These findings showed that such platforms could be considered potential preventive cancer vaccine candidates in the clinic [10]. Therefore, this review summarizes the properties of nanovaccines, the different types of vaccines, various nanocarriers, adjuvants and also discusses the findings of the latest studies in the field of cancer therapy and cancer prevention employing nanovaccines as well as the advantages and disadvantages of this therapeutic approach.

## 2. What Are Nanovaccines Composed of?

Evidence demonstrated that subunit vaccines alone could not protect people against deadly pathogens because the immune activation by these vaccines remained weak, and the duration of protective immunity was also found to be low [9]. To overcome the restrictions of traditional vaccine adjuvants, the use of NP-based delivery vehicles such as virosomes, liposomes, micelles, microemulsions, dendrimers and nanogels could provide a potential approach [9,11]. Based on previous studies, nanovaccines can enhance the delivery of antigens and adjuvants, antigen presentation by APCs, stimulation of innate immune responses, and robust effector T cell responses to kill pathogenic microorganisms and tumor cells with minimum toxicity and adverse effects [12]. Therefore, nanovaccines can be extremely advantageous in creating effective immunotherapeutic formulations for human malignancies [13]. Traditionally, nanovaccines are composed of antigens, adjuvants (molecular or NP-based) and nanocarriers (Figure 1).

### 2.1. Antigens

Tumor antigens are categorized into tumor-specific antigens (TSAs) and tumor-associated antigens (TAAs). TSAs are found on tumor cells but not on non-cancerous cells, while TAAs are found at elevated levels on cancer cells but are also expressed at lower levels on non-cancerous cells [14]. MHC-I can present tumor antigens on the surface of tumor cells to prime T lymphocytes [15]. Endogenous tumor antigens cannot induce significant immune responses based on immunosuppression and immune evasion mechanisms. It has been demonstrated that exogenous tumor-associated antigens could be used in cancer vaccines by inducing antigen-specific immune responses against tumor cells in cancer immunotherapy [16]. TAAs can be expressed by tumor cells in various malignancies as well as normal cells. So far, numerous TAAs have been identified, and some have been used in cancer vaccines [17]. The most important TAAs are HER-2/neu, melanoma-associated antigens-A (MAGE-A), TTK protein kinase (TTK), LAGE-1, and the gene encoding New York’s esophageal squamous cell carcinoma 1 (NY-ESO-1), which is used in cancer vaccines [18,19,20,21].

On the other hand, TSAs or neoantigens that differ from wild-type antigens are only expressed by tumor cells and result from genetic occurrences such as abnormal gene expression and accidental somatic mutations in tumor cells [22,23,24]. Moreover, neoantigens are not subjected to the central tolerance process and can be identified by the immune system as non-self-antigens [25]. Therefore, immunotherapy-based therapies include neoantigen vaccines and neoantigen reactive T cell (NRT) therapy [26,27].

Studies revealed several challenges and limitations related to targeting TAAs, and so far, the clinical outcomes of using TAA cancer vaccines have been almost unsatisfactory [28]. TAA vaccines must overcome acquired and central tolerance and limit the magnitude of induced T cell responses. Clinical studies have also shown that administering TAA vaccines may lead to autoimmunity and on-target off-tumor toxicity [29].

### 2.2. Immunostimulatory Adjuvants

Adjuvants are used as ingredients in some vaccines, creating a robust immune response in people receiving the vaccine [30]. In nanovaccines, immune adjuvants can also induce and guide immune responses against antigens. The immunostimulatory properties of adjuvants are critical for subunit antigens which are inherently weakly immunogenic [31]. Adjuvants can be categorized into two classes based on their mechanisms of action, including vaccine delivery systems and immunostimulatory molecular adjuvants [32]. Emulsions, mineral salts, virosomes, and liposomes are vaccine delivery systems that can induce more efficient antigen presentation to effector immune cells and control antigen release and deposition. Immunostimulatory molecular adjuvants such as stimulator of interferon genes (STING) agonists, Toll-like receptor (TLR) agonists, cytokines, and co-stimulatory ligands can activate APCs, enhancing antigen presentation to T cells and further desirable immune responses [2,33].

As discussed, loading adjuvants and antigens into NPs can protect adjuvants and antigens against degradation by proteases, phosphatases, and nucleases [34]. It was shown that adjuvants could be associated with systemic toxicity characterized by fever, diarrhea, nausea, and lethargy in vaccine receivers [35]. Consequently, the encapsulation of adjuvant and antigen using NPs can protect the organs from this systemic toxicity. Nanoencapsulation can also be employed to induce and stimulate immune responses by slow-releasing antigens [35]. As satisfactory depots, polymeric or gel-like NP platforms can gradually release antigens and adjuvants over a more extended period of time [36]. In terms of nanomaterial-based vaccine adjuvants, aluminum hydroxide, aluminum oxyhydroxide, gold, silver, mesoporous silica, poly lactic-co-glycolic acid (PLGA), chitosan, and DDA liposome have been used in several studies [37]. 

## 3. Nanocarriers

Nanocarriers may be considered the most important part of nanovaccines as they carry subunit vaccines containing antigens and adjuvants for delivery. An extensive list of nanocarriers have been used in cancer vaccines and immunotherapy, which can be classified into three main categories: biogenic, semi-synthetic and synthetic nanocarriers, which are briefly mentioned in this section (Figure 2).

### 3.1. Biogenic Nanocarriers

Biogenic nanocarriers are a group of nanomaterials derived from biological organisms such as biological cells with low toxicity, high potential biocompatibility and high biodegradability. This section presents two widely used and important examples of biogenic nanocarriers, including outer membrane vesicles (OMVs) and exosomes, along with their characteristics.

#### 3.1.1. Outer Membrane Vesicles

Outer membrane vesicles, as bioparticles derived from the outer membrane of Gram-negative bacteria, have a diameter of about 50 to 250 nm, which is an appropriate size for efficient carriers for antigen transfer to lymph nodes and intracellular delivery to APCs. Therefore, OMVs have been paid increasing attention in cancer immunotherapy. In bacteria, OMVs are involved in the trafficking of biochemical signaling that may include RNA, DNA, endotoxins, proteins, and virulence molecules. In addition to acting as carriers, OMVs can also be used as appropriate adjuvants for vaccine development because bacterial OMVs contain immune-stimulating danger signals such as lipopolysaccharide, lipoprotein, and flagellin stimulating TLR4 and TLR5, respectively [38,39]. So far, OMVs have been employed to make bacterial and cancer vaccines. In cancer vaccines, OMVs can be loaded with TLR agonists and anti-tumor cytokines to stimulate prolonged anti-tumor immune responses and eliminate tumor cells with minimal side effects [40]. Engineered OMVs can also be used in the development of cancer vaccines. In this regard, it has recently been shown that the OMV-specific programming of DCs can lead to the maturation of these APCs and the survival of antigen cross-presentation to CD8^+^ T cells [41]. Recently, a study identified tumor antigens on the surface of OMVs by fusing them with cytolysin A protein and then simplified the antigen display procedure using a Plug-and-Display system including the tag/catcher protein pairs. OMVs decorated with different protein catchers can concurrently display multiple, diverse tumor antigens to produce antitumor immune responses in a synergistic fashion. Furthermore, bioengineered OMVs loaded with various tumor antigens can inhibit melanoma-induced lung metastasis and suppress the growth of subcutaneous colorectal cancer. It is possible that the rapid and concurrent display antigens simplify the development of bioengineered OMVs for individualized cancer vaccines [42].

#### 3.1.2. Exosomes

Another group of biogenic carriers are exosomes of about 30 to 150 nm in size that possess a high potential for efficient immunotherapy and vaccine delivery. Exosomes are secreted by a wide range of cells such as APCs, tumor cells, B cells, and T cells and depending on the cellular origin and different pathological conditions, exosomes can suppress or stimulate the immune system. Therefore, exosomes can affect the immunotherapy of tumors or autoimmune diseases. In cancer immunotherapy, tumor-derived exosomes containing MHC/epitope molecular complexes detected by T cell receptors (TCRs) and activated effector T cells are effective. DC-derived exosomes also contain molecules and receptors involved in antigen presentation and T cell activation. Moreover, some studies have shown that exosomes and their contents, such as exosomal microRNAs (miRNAs), can be used as a prognostic tool to determine the stage of the disease, which can be useful in diagnosis and treatment [43]. Previous experience revealed that the development and use of exosome-based nanodrugs and nanovaccines are fraught with many limitations and challenges, such as the cost and time required to manufacture exosomes, particularly on a large clinical scale.

Studies have introduced novel methods for the identification, isolation, and molecular characterization of disease-associated exosomes that can be designed based on the antigenic reactivity of exosomes. In addition, the need for new technologies to identify and introduce specific markers of exosomes and their subgroups, as well as engineering target-guided exosome-like particles can open a new window for using exosomes in the clinic [44]. 

### 3.2. Semi-Biogenic Nanocarriers

Semi-synthetic nanocarriers are composed of synthetic and partially biogenic components. Based on engineering quality, these nanocarriers could have high biocompatibility, low toxicity, and easy and reproducible large-scale manufacturing. Three important types of these semi-synthetic nanocarriers, including virus-like particles (VLPs), endogenous protein-based nanocarriers, and cell membrane-coated nanocarriers, are discussed in this review.

#### 3.2.1. Virus-like Particles

VLPs are noninfectious virus-like particle nanocarriers with no viral genome and are self-assembled from in vitro expressed virus structural proteins. Several expression platforms, such as bacteria, yeast, baculovirus/insect cells (B/IC), plant cells, mammalian and avian cells, and cell-free systems, can be employed for VLP vaccine production [45].

Due to their realizable modification through genetic engineering and the design of antigen expression at the surface, VLPs are a viable option for designing and developing nanovaccines. It was revealed that repetitive antigenic structures engineered on VLP-based nanovaccines can activate the immune system efficiently because VLPs can be captured by APCs, priming robust and durable adaptive immune responses [46]. For instance, in cervical cancer, a vaccine was designed, and the antigens of this vaccine were HPV-16 and HPV-18 L1 VLPs, which are produced in a baculovirus expression vector system. The findings of this study showed that HPV-16 and HPV-18 L1 VLPs could induce immune responses and increase the vaccine’s effectiveness in patients with cervical cancer [47]. Chemical conjugation is another approach to antigen modification on VLPs with bifunctional crosslinkers [48].

#### 3.2.2. Endogenous Protein-Based Nanocarriers

Based on previous studies, albumin is a long half-life endogenous drug carrier used for anticancer drug delivery, radionucleotide, and molecular vaccines [49,50,51]. The delivery of cancer vaccines to lymphoid tissues and APCs is important in post-vaccine immune responses and, in this context, the use of as-assembled protein/drug nanocomplexes with an appropriate size permits effective lymphatic draining and intracellular uptake. Moreover, APCs’ highly expressed neonatal Fc receptor (FcRn) can uptake albumin via endocytosis and simplify the intracellular delivery of nanocomplexes containing vaccine/albumin. The use of albumin is also advantageous compared to synthetic nanomaterials due to its ease of use and high quality based on good manufacturing practices (GMP) and the human body’s high half-life of albumin (20 days). An investigation was undertaken of conjugated molecular vaccines with Evans blue in albumin-binding vaccines (AlbiVax) that self-assembled in vivo from endogenous albumin and AlbiVax. The findings showed that Albumin/AlbiVax could induce peripheral antigen-specific CD8^+^ cytotoxic T cells with a memory of approximately ten times that of incomplete Freund’s adjuvant-emulsifying vaccines in B16F10, MC38, and EG7.OVA tumors [51].

#### 3.2.3. Cell Membrane-Coated Nanocarriers

Another type of nanocarrier is cell membrane camouflage, which is considered a biomimetic platform for drug delivery. After the extraction of the target cell membrane, such as cancer cells, they are coated on nanoparticle surfaces or used as building blocks to form nanocarriers [52,53]. Furthermore, NPs coated by the patient’s tumor cell membrane can carry a wide range of tumor cell membrane antigens and deliver them to APCs for transport to effector lymphocytes [54]. By using this method, it is possible to engineer cell membrane molecules as well as adjuvants and NPs coated with tumor cell membranes with adjuvants, and a targeted ligand can be used effectively as a nanovaccine in the treatment of cancer. It was reported that pH-sensitive liposomes coated with the membrane of macrophages effectively delivered emtansine as an antitumor drug to inhibit lung metastasis in metastatic 4T1 breast cancer cells [55].

### 3.3. Synthetic Nanocarriers

Based on the available studies, a wide range of synthetic nanocarriers are used in cancer nanovaccines and cancer immunotherapy, the most important of which are mentioned in this section.

#### 3.3.1. Liposomes

Liposomes are phospholipid bilayer structures with good biodegradability and are useful in the development of nanovaccines. Studies have shown that liposome-conjugated or liposome-encapsulated antigens amplified antigen-specific CD8^+^ T cells proliferation compared to the same antigen alone [56,57]. In this context, bone marrow-derived DCs (BDMCs) incubated with DOTAP-comprising cationic liposomes increased the expression of inflammatory chemokine genes, the maturation of CD11c+, and upregulated the expression of co-stimulatory molecules such as CD80 and CD86 in vitro and in vivo [58,59].

#### 3.3.2. Polymer Nanoparticles

Polymer NPs such as poly lactic-co-glycolic acid (PLGA)-based NPs have been studied extensively for vaccine delivery in cancer immunotherapy [60]. PLGA has good biodegradability, and in vivo its ester bonds can eventually be broken down into metabolizable monomers of lactic acid and glycolic acid. Moreover, PLGA NPs’ size, stability, and solubility are well adjustable. In addition, to block the formation of copolymers, PLGA is able to couple with polyetherimide or polyethylene glycol (PEG) [2,61]. These block copolymers can aggregate spontaneously into a polymeric micelle, and the micelles can enclose hydrophobic peptide antigens [62]. Evidence suggests that antigen-containing polymeric nanovaccines increase T cell responses more effectively than molecular antigens [63]. It was reported that poly(ethylenimine)-coated PLGA (OVA) NPs induced antigen cross-presentation and strong CD8^+^ cytotoxic T cell-mediated immune responses, and could be employed for efficient anticancer immunotherapy [64].

#### 3.3.3. Inorganic Materials

Inorganic minerals have also been widely studied in the development of nanovaccines [65]. In this context, it was demonstrated that immune cells effectively detect and induce the phagocytosis of inorganic nanocarriers. In the field of cancer treatment, studies in animal models have shown that TAAs conjugated to some mineral NPs can detect target antigens and suppress tumor growth [66,67]. Spherical nucleic acids (SNAs) are also used to deliver molecular vaccines, which are formed in nuclei containing gold NPs (AuNP) and nucleic acids on the surface [68]. The AuNP nucleus allows SNAs to enter cells without the use of transfection reagents and delivery vehicles, and these vaccines can strongly stimulate the immune system to produce antibodies and cellular immune responses [69]. Furthermore, SNAs have the ability to modulate the immune system. An investigation recently found that SNA NPs administration in mice lymphoma models improved doxorubicin accumulation in the TME to promote tumor cells apoptosis and autophagy, activating immunogenic cell death and autophagy-mediated T helper1-type immune responses. Moreover, co-delivered CpG with doxorubicin synergistically stimulated antitumor responses, inhibited tumor growth and prolonged animal survival [69].

### 3.4. Self-Adjuvanted Nanocarriers

An interesting feature of some nanomaterials is that they can act as effective immunoadjuvants and vaccine carriers [70]. Examples of these nanomaterials are chitosan, Al_2_O_3_ NPs and polymethyl methacrylate NPs. These NPs have a high potential to enhance cellular and humoral immune responses and produce a balanced Th1/Th2 response [71,72,73]. However, some NPs may aggravate adverse allergic reactions due to their immunostimulatory effect [74,75]. In a recent study, hyaluronate and trimethyl chitosan recoated superparamagnetic iron oxide nanoparticles loaded with hypoxia-inducible factor-1α-silencing siRNA and E7046 (EP4 antagonist) were used to treat tumor cells, and the outcomes showed that this nanosystem could inhibit cancer cell colony formation, proliferation, migration, invasion, and angiogenesis remarkably [76].

## 4. Types of Nanovaccines

This section evaluates four main types of nanovaccines, including neoantigen nanovaccines, STING agonist-based nanovaccines, artificial APC nanovaccines, and RNA-based nanovaccines (Figure 1).

### 4.1. Neoantigen Nanovaccines

As mentioned earlier, neoantigens are produced following somatic mutations in tumor cells but not in healthy cells [26]. Therefore, neoantigen vaccines based on DNA, mRNA or synthetic peptides are potential therapeutic targets in cancer treatment [23]. Neoantigens can be identified by genomically sequencing cancer cells and normal cells or using an MS-based proteomics analysis, and after the sequencing step, somatic mutations can be identified using in silico methods, and MHC-I/II binding to neo-epitopes is then predicted. Another important step in the study of neoantigens is to determine the immunogenicity of neoantigens in vitro and, finally, the synthesis of target neoantigen peptides [77]. One of the major challenges in the field of neoantigens is their low immunogenicity, and nanovaccines can, to some extent, improve vaccine delivery and thus the immunogenicity of neoantigens. Studies demonstrated that synthetic high-density lipoprotein nanodiscs can be used as clinically safe and scalable nanomaterials to facilitate the delivery of peptide neo-antigens through disulfide conjugation, and cholesterol-modified adjuvant therapy to draining lymph nodes can be effective in treating cancer by inducing antigen-specific CD8^+^ T cell responses [78]. The administration of neoantigen nanovaccines to mice models of melanoma showed that the mean recurrence time was prolonged from 11 to 16 days, and the median survival time was extended to about eight days compared with the control group. In addition, the frequency of neoantigen-specific T cells increased to 10-fold of free vaccines, thereby increasing the levels of TNF-α and IFN-γ. In the nanovaccine group, the antitumor action of spleen lymphocytes was markedly stronger than in other studied groups. However, the infiltration of immunosuppressive cells in the TME, as well as the expression of inhibitory molecules, presents a challenge in this method, which can be overcome by using immune checkpoint blockers [79].

### 4.2. STING Agonist-Based Nanovaccines

STING is an endoplasmic reticulum-associated signaling molecule involved in the transcriptional regulation of several immune system-related genes, thus playing an important role in the innate immune responses against numerous bacterial and viral pathogens [80]. Evidence suggests that STING increases the expression of type I interferons (IFNs) and proinflammatory cytokines following cytolytic DNA sensing by cytosolic cyclic GMP–AMP synthase (cGAS) in terms of inflammatory pathway initiation and pathogen clearance [80]. Moreover, following the administration of DNA vaccines, adaptive immunity can be induced via STING-dependent signaling [81]. In the field of cancer studies, it was revealed that CD8α^+^ DCs can release type I IFN via the STING pathway, resulting in antigen cross-presentation and the priming of CD8^+^ T cells [82]. Furthermore, B cells and other CD11b^+^ tumor-infiltrating host APCs can recognize tumor-derived STING-activating components, releasing STING-mediated type I IFN induced by leukocytes and cytotoxic NK cells’ priming for tumor cell elimination [83]. In addition, the STING pathway contributes to the induction of natural anti-tumor T cell responses or radiotherapy-induced T cell responses [84]. Studies revealed that by encapsulating cyclic dinucleotides (CDNs), NPs could enhance cytosolic CDN delivery and induce immune responses. For instance, cyclic di-GMP encapsulation into PEGylated lipid NPs as a cancer nanovaccine could significantly activate CD4^+^ and CD8^+^ T cell responses compared to CDNs alone [85]. It was reported that the intratumoral administration of STING-activating PC7A nanovaccine in mice models of melanoma could increase antigen-specific CD8^+^ cytotoxic T cell infiltration in the TME. Moreover, the accumulation of STING-activating PC7A nanovaccine in tumors upregulated the expression of CXCL9 in myeloid cells, enhancing the recruitment of IFNγ-expressing CD8^+^ T cells from the periphery into the TME [86].

The outcomes outlined above indicate that employing STING agonist-based nanovaccines may enhance the CDNs’ bioavailability and therapeutic effectiveness due to the induction of STING signaling and anti-tumor immune responses.

### 4.3. Artificial APCs

Antigen-presenting cells, including DCs, macrophages, and B cells, are responsible for antigen capture, processing, and presenting via MHC-II. However, several other cells, such as keratinocytes, fibroblasts, and endothelial cells, can, in some cases, present antigens to lymphocytes to trigger adaptive immune responses [87,88,89,90,91]. Among these APCs, DCs have received the most attention, and autologous DC-based cancer vaccines have achieved great success in the treatment of cancer, so much so that in 2010, a vaccine called PROVENGE (Sipuleucel-T) was approved by the Food and Drug Administration for the treatment of prostate cancer [92]. However, this therapeutic approach has various setbacks, including its time-consuming and costly process, as well as the limited availability of autologous cellular resources. Therefore, the development of artificial APCs was suggested to address these challenges. aAPCs are synthetic APCs composed of a cognate antigenic peptide presented by MHC molecules for binding to TCR and co-stimulatory molecules for binding to related receptors at the immunological synapse, thereby activating T cells [93]. One of the advantages of using aAPCs instead of natural APCs is the defined composition along with manageable aAPCs signals.

On the other hand, aAPCs can be produced on a large scale and used as ready-to-administer vaccines. It has been shown that various parameters such as the size and shape of aAPCs may affect T cell activation, and these physicochemical properties of aAPCs should be further engineered to optimize the immune system and treatment [94]. A study in this area transformed tumor cells into aAPCs by infecting them with a herpes simplex virus 1-based oncolytic virus encoding IL-12 and TNF superfamily member 4 (TNFSF4 or OX40L) to induce stimulatory signals for the maximum activation of effector T cells. The aAPCs could express APC-associated biomarkers and induce the activation of antigen-specific T cells and their killing ability in co-cultures with tumor infiltrated lymphocytes (TILs) in vitro. Furthermore, combining OV-OX40L/IL-12 and TIL therapy induced complete tumor regression in tumor mice models and elicited antitumor immune memory. Additionally, this combination therapy reprogrammed TME components, such as tumor-associated macrophages (TAMs), to the M1 phenotype [95].

### 4.4. RNA-Based Nanovaccines

Previous evidence and studies in cancer treatment show that these therapeutic approaches can be useful in this field due to the wide range of RNA-based therapies. Using positive regulatory RNAs (such as mRNA) or inhibitors such as siRNA and microRNA can alter the expression of target proteins [96]. Like DNA delivery, mRNA delivery aims to positively and purposefully regulate the expression of the studied proteins, with the difference that mRNA drugs do not contain the risk of insertional mutations and possess more consistent and predictable protein expression kinetics. These drugs are also easier to synthesize in vitro [97]. Another advantage of mRNA over DNA is its greater transfection efficiency, especially in immune cells [98,99]. Despite these advantages, the use of naked RNA in vitro faces certain limitations such as poor chemical stability, short half-life, and degradation by nucleases [98,100]. Therefore, using NPs can help to stabilize RNA molecules and create a targeted delivery system [101]. These NPs encapsulate RNAs to protect them against enzymatic degradation and clearance by the immune system.

On the other hand, nanotechnology facilitates the delivery and penetration of RNA to infiltrated immune cells at the tumor site [102,103]. In this context, NP-based platforms for RNA delivery include lipid-based nanostructures, polymer-based nanomaterials, inorganic NPs, and bio-inspired nano-vehicles [96]. In vivo mRNA delivery efficiency was significantly enhanced by nanovector formulations; nonetheless, the efficacy of mRNA vaccines remains unsatisfactory. A lipoplex-based epitope-encoding mRNA nanovaccine (mRNA@lipoplex) was designed and fabricated to improve mRNA delivery in APCs, activate specific T cell responses, and induce strong antitumor immunity. The findings showed that this nanovaccine significantly improved mRNA capture by DCs and amplified the activation of effector T cells, which was confirmed by the release of IFN-γ and IL-2 in vitro. Furthermore, NPs in this vaccine enhanced the expression of IL-12 by inducing the nuclear factor kappa-light-chain-enhancer of the activated B cells (NFKB) pathway in DCs. The mRNA@lipoplex nanovaccine activated CD8^+^ T cell and prevented tumor progression in mice models of melanoma and colon cancers. Interestingly, the mRNA@lipoplex nanovaccine extended the studied animals’ survival and produced long-term antitumor memory [104]. Therefore, these nanosystems can induce antitumor responses in addition to maintaining mRNA stability and its effective delivery to the tumor site synergistically.

## 5. Nanovaccines in Cancer Therapy

Cancer vaccines that include TAAs can induce an effective anti-tumor immune response via APCs, such as DCs and macrophages, and have revealed promising cancer prevention abilities and therapeutic potential. Nevertheless, ambiguous immunization processes and poor anticancer effectiveness previously limited cancer vaccine application [105]. To overcome these limitations and challenges, nanotechnology-based techniques can increase the effectiveness and stability of anti-tumor responses to cancer vaccines. This section discusses the outcomes of the latest studies (from after 2020 to now) in this field (Table 1).

Using nanomedicines in cancer vaccines offers valuable opportunities to improve the efficacy of these vaccines and cancer immunotherapy. Based on the available evidence, various nanoplatforms for delivering cellular, molecular, and intracellular vaccines have been studied to target lymphoid cells and various tissues. The results generally demonstrate the robustness and durability of antitumor immune responses induced by these platforms. Moreover, at the same time, undesirable side effects were minimized following this treatment strategy [2].

An investigation of orthotopic melanoma models reported that a multifunctional nanovaccine termed OMPN comprising polydopamine, ovalbumin (OVA), and MnO2 was prepared by employing a facile one-pot method and could inhibit tumor growth and liver metastasis in a melanoma mice model. Furthermore, MRI tracking showed that the locomotion and migration of DCs significantly increased in the inguinal lymph node following vaccination, signifying effective DC activation and anti-tumor immune response. An evaluation of the TME following OMPN treatment showed that the infiltration of CD3^+^ and CD8^+^ cells, as well as release of IFN-γ even after exposure to the laser, significantly increased. Moreover, in the OMPN + laser group, the polarization of TAMs from the M2 to M1 phenotype was better amplified than in other groups. The findings were confirmed by decreased IL-10 and increased IL-12p40 levels in the TME following treatment with OMPN + laser. The authors suggested that OMPN could be used as an effective and MRI-trackable nanovaccine for treating melanoma [105].

Another personalized nanovaccine using cationic fluoropolymer and OVA (F-PEI/OVA) for post-surgical cancer immunotherapy showed that F-PEI/OVA could induce the maturation of DCs via the TLR4-mediated signaling pathway and improve antigen transportation into the cytosol of DCs, resulting in the enhancement of antigen cross-presentation by these APCs and ovalbumin-expressing B16-OVA melanoma inhibition. Moreover, the finding showed that in melanoma and breast subcutaneous tumor models, the combination of fluoropolymer with resected autologous isolated cell membranes and checkpoint blockade therapy could synergistically inhibit post-surgical tumor recurrence and metastases. In addition, a robust immune memory against tumor rechallenge was detected in tumor models. F_13_-PEI (fluorine content  =  20.49%)/OVA-immunized mice demonstrated a high survival rate, with 37.5% surviving for up to two months, indicating the potential of F_13_-PEI/OVA as an effective cancer nanovaccine. Therefore, the lipophobic and hydrophobic chemical properties of fluoroalkane chains could suggest them to be a unique antigen carrier and immune adjuvant for nanovaccines’ construction compared with other vaccines using conventional adjuvants, such as alum and CpG [106].

L-arginine (LA)-loaded black mesoporous titania (BMT), a multifunctional nanovaccine, was used to improve anti-tumor therapeutic efficacy and inhibit tumor metastasis. BMT in this system was employed as an acoustic sensitizer for sonodynamic therapy, and LA was used as exogenous nitric oxide supplementation for gas therapy. It was reported that the ultrasound simultaneously stimulated LA and BMT to produce singlet oxygen (^1^O_2_) and NO gas in the TME. Fascinatingly, ultrasound-excited BMT producing ^1^O_2_ can enhance LA oxidation to produce more nitric oxide. The high concentration of nitric oxide and ^1^O_2_ in tumor cells can greatly increase the intracellular oxidative stress levels and dsDNA breaks, inducing tumor cell apoptosis. Ultrasound-excited BMT@LA significantly induced DC maturation in vitro. Furthermore, an analysis of DC suspensions following treatment with US-excited BMT@LA demonstrated that the levels of IL-6 and TNF-α significantly increased, promoting antitumor immune responses. The findings also demonstrated that the combination therapy using the ultrasound-excited BMT@LA nanovaccine and immune checkpoint blockers (anti-programmed death-ligand 1 [PD-L1] antibody) could induce a robust anti-tumor immune response by increasing the infiltration of CD8^+^ T cells and releasing TNF-α, IL-6, 1L-12p70 cytokines, resulting in primary tumor cells killing and inhibiting metastasis of tumor cells [107].

A bi-adjuvant neoantigen nanovaccine (banNV) with the capability of co-delivering of a peptide neoantigen (Adpgk) with two adjuvants, including R848 (TLR-7/8 agonist) and CpG (TLR9) agonist, was fabricated for effective colorectal cancer immunotherapy. The banNV nanovaccines were prepared by a nano-templated synthesis of concatemer CpG, cationic polypeptides for nano-condensation, and then physical loading with hydrophobic Adpgk and R848. This study reported that banNV enhanced vaccine uptake by APCs and induced the expression of CD80, CD86, and CD40, as well as the release of IL-6, IL-12, and TNF-α by APCs. The mean tumor progression rate of the treated mice with banNV was 3.65. Following combination therapy, the findings showed that banNVs could sensitize PD-1 receptors on T cells’ surfaces. In addition, the combination of banNVs and anti-PD-1 inhibited tumor growth and increased the survival rate to 70%, while this value for banNV monotherapy was 40% in colorectal cancer models. This study also stated that the anticancer effect of banNV is closely related to the existence of CD8^+^ T cells [108]. Therefore, using nanovaccines may increase the success of treatment with anti-checkpoint blockers in some cases by increasing the sensitivity of cells to treatment. Together, these findings proposed the possibility of using banNVs to potentiate the cancer neoantigens’ immunogenicity for personalized combination immunotherapy in cancer.

A previous investigation provided a novel approach for a clinical personalized anti-tumor vaccine utilizing an R837-loaded PLGA nanovaccine coated with a calcinetin (cancer cell membrane antigen) co-delivering Luc-4T1 tumor cell membrane antigen along with R837 adjuvant. Following vaccination, a personalized anti-tumor immune response was induced. The exposed calcinetin on the surface of the nanovaccine improved the antigen uptake of DCs, enhanced the effectiveness of anti-tumor responses and activated immune memory cells for long-term protection [109].

It was revealed that immune checkpoint blockade therapy is a novel therapeutic approach for treating solid tumors. Nevertheless, the insufficient infiltration of effector T cells in the hypoxic TME limits the anticancer efficacy of immune checkpoint blockers. To overcome this challenge, a sonodynamic therapy nanovaccine integration platform fabricated by binding CpG adjuvant (TLR9 agonist) with manganese porphyrin-based metal-organic frameworks (Mn-MOF) coated by OVA-overexpressing melanoma B16 cells termed cMn-MOF@CM was designed for potentiating anti-PD-1 antibodies in the treatment of malignant melanoma. The findings of this investigation demonstrated that the durability of cMn-MOF@CM was high in circulation and could efficiently improve tumor cells’ targeting.

Moreover, the cMn-MOF@CM could relieve tumor hypoxia, generate strong sonodynamic therapy effects, and induce immunogenic cell death. In addition, a powerful and specific anti-tumor immune response was detected following vaccination due to DC maturation and T cell activation because incubated BDMCs with the cMn-MOF@CM-treated group upon ultrasound irradiation exhibited robust CD80, CD86 and MHCII expression. The frequency of cytotoxic T cell subsets, including CD3^+^ CD8^+^ granzyme B^+^, CD3^+^ CD8^+^ IFN-γ^+^, CD3^+^ CD8^+^ TNF-α^+^, and CD3^+^ CD8^+^ IL-2^+^ T cells significantly increased following treatment with cMn-MOF@CM and ultrasound irradiation. Notably, combining anti-PD-1 antibody and cMn-MOF@CM with ultrasound irradiation showed a long-term systemic anti-tumor immune response, confirmed by increasing the frequency of matured DCs, as well as the infiltration of activated CD8^+^ CD69^+^, CD8^+^ granzyme B^+^, CD8^+^ IFN-γ^+^ and OVA-specific CD8^+^ T cells preventing tumor growth and recurrence [110].

Employing a cancer nanovaccine composed of antigenic peptide, CpG oligodeoxynucleotides and cationic polymer NP significantly triggered DCs’ maturation and induced strong vaccine-specific T cell immune responses. The findings showed that the number of mature CD86^+^ CD11c^+^ DCs increased in the draining lymph nodes from 6.05% for blank mice and 10.8% for untreated tumor-bearing mice to 16.5% for PECT-Cur NPs-treated tumor-bearing mice. Additionally, combining the mentioned nanovaccine with thermo-responsive, curcumin-loaded polymer NPs (a self-assembled hydrogel) in 4T1 models augmented the systemic host T cell immune responses by improving CD8^+^ T cell infiltration at the site of the tumor, attenuating the recurrence of local tumor, and inhibiting metastasis to the lungs [111]. Measuring the weight of relapsed tumors (resected on the 14th day) revealed that the local tumor burden following the combination therapy with PECT-Cur NPs + nanovaccine was diminished by about 85% (0.338 g) compared with PECT (2.495 g) and PBS (2.026 g) groups. These findings suggest that the combination of nanomedicine and nanovaccines can be used as an appropriate post-surgical treatment option, although further studies are needed to evaluate the synergistic effect of this combination therapy.

In order to increase the efficiency of immune checkpoint blockers, another combination therapy was suggested. This study employed natural polycationic protamine (PRT) to carry the unmethylated CpG adjuvant and OVA antigen through appropriate chemical bench-free “green” preparation to preserve the immunological activities of adjuvants and antigens. The finding showed that utilizing polycationic PRT improves PRT/CpG/OVA nanovaccine delivery and enhances DCs’ antigen uptake. In addition, PRT/CpG/OVA nanovaccine stimulated BMDC maturation by inducing the expression of CD80 and CD86, as well as IL-6 and TNF-α secretion. A combination of the anti-PD-1 antibody and the designed nanovaccine was used to inhibit the tumor immune escape and increase tumoricidal activity by enhancing tumor-specific T cell infiltration and the release of TNF-α in the TME [112].

Despite the positive outcomes obtained from such studies, why is cancer treatment still difficult? Due to the complexity of the TME and the different signals it contains, the proper and effective delivery of the antigen to the DCs or blocking of the expressed inhibitory molecules is not sufficient for treatment.

Evidence demonstrates that MDSCs contribute to the diminished overall response to an immune checkpoint blockade. In this regard, an investigation of a colon tumor model showed a TME responsive nanoprodrug (FIT NPs) fabricated for the co-delivery of indocyanine green (ICG) photosensitizer and tadalafil concurrently targeted MDSCs in the TME and intensified tumor immunogenicity. The findings showed a strong therapeutic efficacy by stimulating immunogenic cell death and ameliorating MDSC immunosuppressive activity to enhance photothermal immunotherapy. In the FIT + laser group, the late apoptosis/necrosis rate of MDSCs was reported as 41.0% in vitro. Moreover, FIT + L could markedly suppress tumor growth without changing the studied mice’s weight. The authors suggested that these occurrences lead to DCs’ maturation and T cell activation, strengthening the anti-tumor immune response and immune checkpoint blockade efficacy [113].

Moreover, a small lipid NP-based nanovaccine platform (OVA_PEP_-SLNP@CpG) was constructed by cationic cholesterol derivative and biocompatible phospholipids, and the findings following nanovaccine administration showed that this platform could induce efficient anti-tumor immune responses in therapeutic E.G7 and prophylactic tumor models. This study showed that the frequency of f CD11c^+^MHC II^high^ mature DCs and the expression of co-stimulatory molecules such as CD80 and CD86 increased following the treatment by cross-priming CD8^+^ T cells against the target antigen in vitro. Additionally, it was shown that combining OVAPEP-SLNP@CpG with anti-PD-1 antibody increased the effectiveness of immunotherapy compared to using OVA_PEP_-SLNP@CpG alone. Following the combination therapy in the tumor mouse model, the results showed that the number of PD-1^+^ CD8^+^, CD8^+^ granzyme B^+^, and CD8^+^ IFN-γ^+^ T cells significantly increased, and finally, the size and weight of the tumor mass markedly reduced [114]. Overall, the results of this study showed that although combination therapy with nanovaccines and immune checkpoint blockers can generally increase the effectiveness of immunotherapy; nevertheless, this event depends on various factors, including time and therapy sequences.

A study used a proton-driven nano transformer-based vaccine containing a polymer–peptide conjugate-based nanotransformer and loaded antigen without adjuvant (NTV). The findings demonstrated that this nanotransformer-based vaccine induces a robust anti-tumor immune response without significant systemic toxicity. One interesting characteristic of this nanotransformer-based vaccine is that in the endosomal environment with high acidity conditions, the morphology of the vaccine could change from nanospheres into nanosheets, resulting in disruption to the endosomal membrane directing antigen delivery into the cytoplasm. In addition, following the re-assembling of the nanosheets, anti-tumor immune responses could be promoted by specific inflammation pathways’ activation. Further studies on the human papillomavirus-E6/E7 and B16F10-OVA and tumor mice models showed that the nanotransformer-based vaccine could efficiently inhibit tumor growth. Furthermore, combining NTV with anti-PD-L1 antibody increased the infiltration of CD8^+^ T cells in the tumor and reduced the accumulation of immunosuppressive cells such as Tregs, eliminated tumor cells and prolonged the survival time of the B16F10 model animals. Furthermore, approximately half of the studied mice’s tumors completely regressed [115].

As discussed, neoantigen-based cancer nanovaccines are helpful therapeutic options for the promotion of CD8^+^ T cell responses. In this context, an acid-responsive polymeric nanovaccine was constructed to activate the STING pathway and enhance cancer immunotherapy—the nanovaccines were composed of neoantigen and an acid-activatable polymeric conjugate DMXAA (STING agonist) in a nanoplatform. The outcomes demonstrated that nanovaccines accumulated at the lymph nodes to induce neoantigen uptake by DCs. Additionally, the STING pathway is activated in DCs by the STING agonist to stimulate the secretion of IFN-β and boost neoantigen-specific T-cell priming. Moreover, the nanovaccine significantly reserved tumor growth in B16-OVA melanoma and 4T1 breast tumor mice models. The combination of immunotherapy with anti-PD-L1 antibody and nanovaccines revealed that anti-tumor immune responses were enhanced in a 4T1 breast cancer model [116]. In several other recent studies, it was revealed that the collaborative approaches using nano-based cancer vaccines and immune checkpoint blockers could improve the effectiveness of immunotherapy [122,123,124].

Self-adjuvanted nanovaccines are considered a safe, simple, and cost-effective approach to boosting neoantigen-based cancer immunotherapy. A study designed and fabricated a self-adjuvanted nanovaccine using polymer NPs, and a neoantigen that could activate molecules, inhibit tumor growth and extend the survival time of the studied animals in B16-F10 and colon carcinoma 26 (CT26) tumor mice models. The mechanism of action is the same as that of inducing the maturation of DCs, increasing the infiltration of DCs in the lymph nodes, and finally activating the CD8^+^ T cells to eliminate tumor cells expressing the target antigen used in the vaccine. However, these vaccines are different because self-adjuvant nanovaccines can have abscopal effects in B16-F10 and CT26 tumors and do not require adjuvant or immune checkpoint blockers. Together, these findings can help to address some challenges facing nanovaccines, including the cost and complexity of treatment protocols [117]. Another liposome-encapsulated minimalist nanovaccine (LrTL) composing recombinant protein of trichosanthin (adjuvant) legumain peptide (antigen) was designed, and this study showed that the LrTL could induce a powerful CD8^+^ T cell response by activating DCs that enhance the release of IFN-γ, TNF-α, and IL-12 as immunostimulatory cytokines. Moreover, the nanovaccine could target and eliminate TAMs, inhibiting immunosuppressive effects and TME remodeling. Additionally, in several tumor models, including Lewis’s lung cancer (LLC), B16-F10, intracranial LLC xenograft, and CT-26 colon cancer, the LrTL had powerful anti-tumor activity [118].

Fascinatingly, nanotechnology-based photothermal therapy was successful in cancer therapy because studies in this field reported that serum exosomes (hEX) obtained from hyperthermia-treated tumor-bearing mice exhibited an array of TAAs and robust immunoregulatory capabilities in stimulating the differentiation and maturation of DCs. Black phosphorus quantum dots fabricated a cancer nanovaccine named hEX@BP with exosomes (hEX) encapsulation for treating murine subcutaneous lung cancer models. The hEX@BP and photothermal therapy presented long-term photothermal therapy performance along with tumor temperature elevation in vivo. Moreover, the infiltration of effector T cells into the TME increased following the combination therapy [119]. The results also showed that the combination therapy could increase the survival rate by 80% of the mice at day 40 after tumor inoculation.

Another nanoplatform using mesoporous silica NPs (MSNs) as a vector, which integrated OVA, photothermal agent polydopamine (PDA), and antigen-release promoter ammonium bicarbonate (ABC), was designed and constructed for melanoma photothermal immunotherapy. The MSNs-ABC@PDA-OVA nanovaccine with powerful photothermal properties was able to eliminate primary tumors excellently. The findings demonstrated that upon irradiation by laser light, the MSNs-ABC@PDA-OVA nanovaccine recognized released antigens, enhancing DCs maturation and activation, as confirmed by increased CD40, CD80, and CD86 expression in vitro. These occurrences can facilitate the migration of the activated DCs to tumor-draining lymph nodes and stimulate strong anti-tumor immune responses. Notably, a single dose of MSNs-ABC@PDA-OVA combined with a single round of photothermal therapy effectively eliminated melanoma tumor cells by inducing CD8^+^ T cells infiltration, releasing IFN-γ and TNF-α, as well as creating a robust immunological memory to prevent tumor recurrence and lung metastasis [120]. Other biocompatible photothermal therapy agents, such as Gold nanorods (AuNRs), are effective in cancer immunotherapy [125]. 

As mentioned before, RNA-based cancer nanovaccines such as mRNA vaccines are a satisfactory candidate for cancer immunotherapy. These types of vaccines can safely encode TAAs. In this context, an injectable hydrogel that can generate OVA mRNA, composed of polyethylenimine (PEI) and graphene oxide (GO), loaded with R848-laden, was constructed. The released nanovaccines could protect mRNA’s degradation and facilitate targeted delivery to lymph nodes. The findings revealed that a single dose of this transformable hydrogel could remarkably upsurge the frequency of specific CD8^+^ IFN-γ^+^ T cells and release TNF-α, thereby inhibiting tumor growth and generating antigen-specific antibodies that prevent metastasis occurrence [121].

## 6. Challenges of Nanovaccines for Cancer Therapy

Evidence demonstrated that immunotherapy-based approaches are limited in solid tumors. This is because the immunogenicity of tumors is poor and anti-tumor T cell-mediated immune responses do not possess the necessary efficiency, resulting in a low response rate for patients with cancer [126,127]. Among the immunotherapy-based approaches, cancer vaccines also encounter several challenges, and their therapeutic efficiency is strongly influenced by factors such as tumor heterogeneity, low immunogenicity, poor in vivo delivery, tumor immune escape, high treatment costs, complications, and low persistence in the blood circulation [112,117]. To address these limitations, nanovaccines were designed and demonstrated promising efficiency in treating cancer; however, the design, manufacture, and administration of these nanovaccines also have limitations [128] (Figure 3). For instance, in RNA-based nanovaccines, the inherent RNA instability and translation efficiency to proteins are the main limitations [121].

Moreover, tumor heterogeneity significantly hampers the development of nanovaccines. It has been shown that improving cancer immunotherapy efficiency by combination immunotherapy using nanodrugs for improving tumor immunogenicity and nanovaccines for enhancing the anti-tumor T cell responses may be a promising avenue in cancer therapy [111]. Another challenge with nanovaccines is their impact on effector immune cells. In this regard, an investigation reported that nanovaccines (e.g., OVA_PEP_-SLNP @CpG) could induce T cell exhaustion by upregulating PD-L1 expression, resulting in tumor recurrence [114]. NP-based combination therapy using immunotherapy and photothermal therapy could be an attractive approach to increase tumor ablation and improve anti-tumor immune responses.

Nevertheless, a lengthy and complicated treatment process and a poor immune response often hinder this therapeutic method. It has been suggested that checkpoint blockers should be incorporated into the treatment protocol to improve the therapeutic effect [120]. Finally, it should be kept in mind that the effective delivery of antigens by nanovaccines is not always indicative of effective anti-tumor responses by antigen-specific CD8^+^ T cells because it has been shown that after the administration of nanovaccines, despite the effective responses of antigen-specific CD8^+^ T cells, the cytotoxic function of these lymphocytes have either reduced or diminished, which may be due to the escape of tumor cells caused by the reduced expression of MHC molecules or the presence of other inhibitory signals in the TME [108,129]. 

## 7. Concluding Remarks

It seems that cancer nanovaccines can effectively treat cancer through the proper delivery of tumor antigens to APCs, which leads to the maturation and activation of these cells and increases the infiltration of anti-tumor function CD8^+^ T cells. These nanovaccines can also be used in combination with other therapeutic approaches such as radiotherapy, immunotherapy, and chemotherapy, and by synergizing anti-tumor responses, they can eliminate the tumor, inhibit metastasis, and increase survival. According to this review of the latest studies on nanovaccines in cancer therapy, it is essential to comprehensively elucidate the mechanism of action of this nanovaccine [113]. Correspondingly, nanovaccine safety should be studied systematically. Moreover, the effectiveness of the nanovaccine should be further verified in different tumor models and clinical studies. 

## Authors Contributions

K.J.: Conception, design and inviting co-authors to participate. X.F. and H.L.: Writing original manuscript draft. K.J., J.Q. and D.G.: Review and editing of manuscript, which were critically important for intellectual content, and provided comments and feedback for the scientific contents of the manuscript. All authors have read and agreed to the published version of the manuscript.

## Figures and Tables

**Figure 1 cancers-14-03842-f001:**
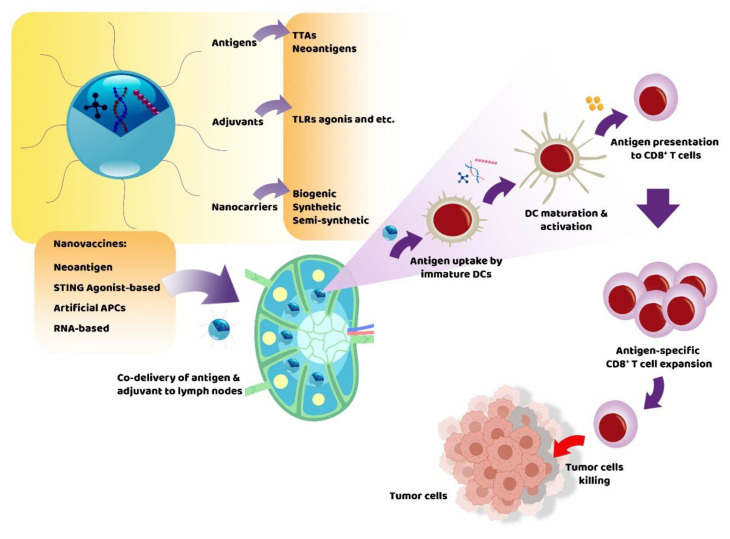
Nanovaccines in the treatment of cancer. The general structure of nanovaccines, their types, and the mechanism of action of this type of vaccine are shown. After administration of nanovaccine and delivery of antigen and adjuvant to lymphoid tissues, antigens are uptake by DCs, resulting in DCs maturation and activation. After this stage, the matured DCs present the antigens to the CD8^+^ T cells through the MHC molecules and cause T cell expansion. Finally, antigen-specific T cells invade tumor cells in the TME and kill them. APC: antigen-presenting cell; DC: dendritic cell; TAAs: tumor-associated antigens; TLR: Toll-like receptor.

**Figure 2 cancers-14-03842-f002:**
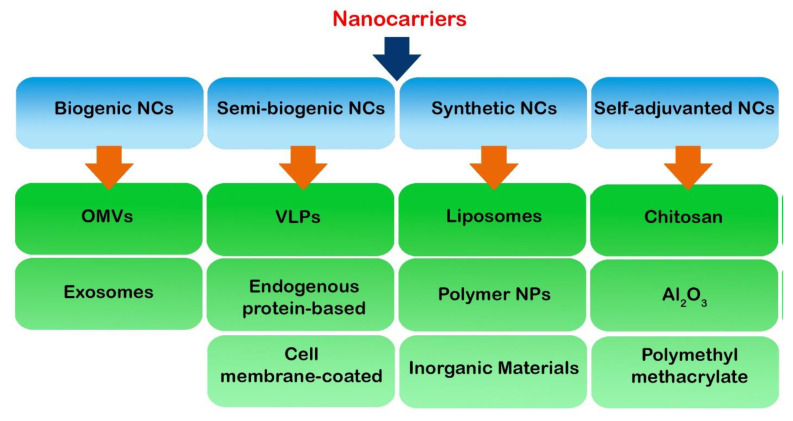
Three main nanocarriers are shown, including biogenic, semi-synthetic and synthetic NCs. NP: nanoparticle; NC: nanocarrier; OMV: outer membrane vesicles; VLP: virus-like particle.

**Figure 3 cancers-14-03842-f003:**
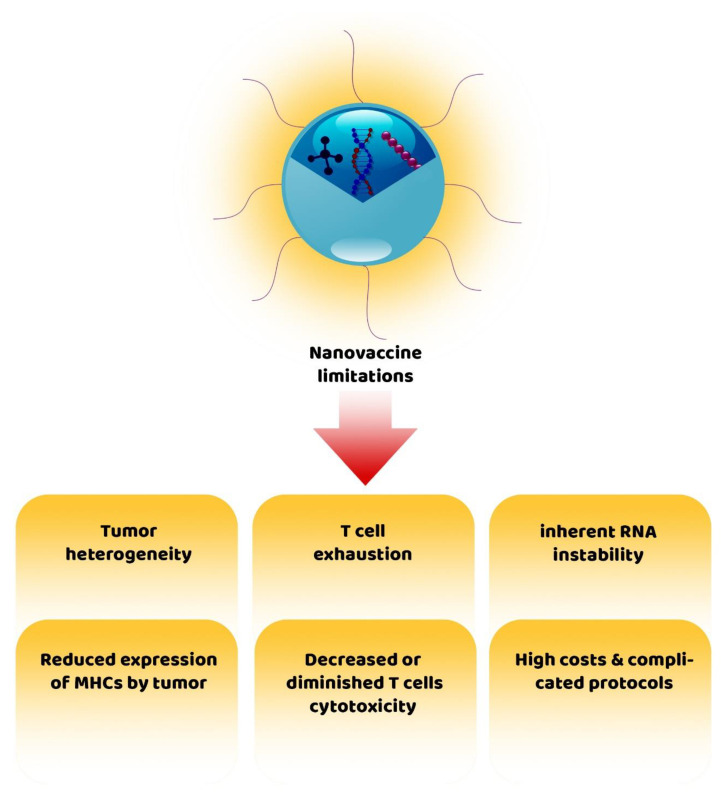
Challenges and limitations of cancer treatment with nanovaccines.

**Table 1 cancers-14-03842-t001:** The latest nanovaccines in cancer therapy.

Nanovaccine	Type of Study/Cancer	Mechanism of Action	Outcomes	Ref
**OMPN** (OVA, MnO2, and polydopamine)	In vitro/Animal model/orthotopic melanoma	Increasing the locomotion and migration of DCs in the inguinal lymph node Induction of anti-tumor immune responses	Inhibiting tumor growth and liver metastasis Increasing CD3^+^CD8^+^ T cells Inducing polarization of M2 to M1 macrophages phenotype Increasing IFN-γ and IL-12p40 Decreasing IL-10	[105]
**F-PEI/OVA** (OVA, fluoropolymer)	In vitro/Animal model/orthotopic melanoma and breast cancer	Inducing DCs maturation via the TLR4-mediated signaling pathway Improving antigen transportation into the cytosol of DCs Enhancing antigen cross-presentation	Inhibiting post-surgical tumor recurrence and metastases Prolonging survival rate for up to 60 days	[106]
**BMT@LA** (L-arginine, black mesoporous titania)	Animal model/bilateral U14 tumor model	Producing singlet oxygen (^1^O_2_) and nitric oxide gas Increasing the intracellular oxidative stress levels Increasing dsDNA breaks Inducing tumor cell apoptosis	Inhibiting tumor growth and metastasis Increasing the infiltration of CD8^+^ T cells Releasing TNF-α, IL-6, 1L-12p70 cytokines	[107]
**banNV** (Adpgk neoantigen, R848, CpG)	In vitro/Animal model/ MC38 colorectal cancer cell	Sensitizing PD-1 receptor on the surface of T cells Increasing vaccine uptake by APCs	Enhancing the expression of CD80, CD86, and CD40 Increasing IL-6, IL-12, and TNF-α by APCs Inhibiting tumor growth Increasing survival rate to 70%	[108]
**PLGA nanovaccine** (calcinetin, R837)	In vitro/Animal model Luc-4T1 cells	Co-delivering calcinetin-expressed cancer cell membrane antigen and R837 adjuvant	Inducing a personalized anti-tumor immune response Improving the antigen uptake of DCs Enhancing the effectiveness of anti-tumor responses Activating immune memory cells to provide long-term protection	[109]
**cMn-MOF@CM** Mn-MOF, CpG, OVA	In vitro/Animal model Melanoma B16	Promoting DC maturation and T cell activation Relieving tumor hypoxia Inducing immunologic cell death Inducing long-term immunological memory function to	Prolonged blood circulation and enhanced tumor targeting Increasing the expression of CD80, CD86 and MHCII Increasing the frequency of cytotoxic T cell subsets, including CD3+ CD8+ granzyme B+, CD3+ CD8+ IFN-γ+, CD3+ CD8+ TNF-α+, and CD3+ CD8+ IL-2+ T cells	[110]
(Antigenic peptide, CpG oligodeoxynucleotides and cationic polymer NP)	In vitro/Animal model Breast carcinoma 4T1 cells	Increasing mature CD86+ CD11c+ DCs Inducing strong vaccine-specific T cell immune responses Enhancing CD8+ T cell infiltration at the site of tumor, attenuating the recurrence of local tumor and inhibiting metastasis to the lungs	Amplifying the systemic host T cell immune responses Attenuating recurrence of local tumor Reducing tumor weight following combination therapy with PECT-Cur NPs + Nanovaccine Inhibiting metastasis to the lungs	[111]
**PCO** (PRT/CpG/OVA)	In vitro/Animal model BDMCs, B16 melanoma cells	The polycationic PRT lead to improve PRT/CpG/OVA nanovaccine delivery Enhancing DCs’ antigen uptake and maturation	Increasing the efficiency of immune checkpoint blockers Combining the anti-PD-1 antibody and the PRT/CpG/OVA nanovaccine leads to inhibition of tumor immune escape Increasing the tumoricidal activity Improving tumor-specific T cells infiltration Increasing the expression of CD80 and CD86 on BDMCs Inducing the release of IL-6 and TNF-α	[112]
**Nanoprodrug** (FIT NPs, tadalafil ICG photosensitizer)	In vitro/Animal model CT-26 cells/ Colon cancer	Targeting MDSCs in the TME and intensifying tumor immunogenicity DCs maturation and T cells activation Increasing tumor immunogenicity	Stimulating immunogenic cell death by ICG photosensitizer Ameliorating MDSCs’ immunosuppressive activity by tadalafil for enhancing the photothermal immunotherapy Strengthening anti-tumor immune response and immune checkpoint blockade efficacy Increasing the number of PD-1+ CD8+, CD8+ granzyme B+, and CD8+ IFN-γ+ T cells in the tumor Reducing the size and weight of the tumor	[113]
**OVAPEP-SLNP@CpG** (Small lipid nanoparticle, CpG, OVA)	In vitro/Animal model Prophylactic and therapeutic E.G7 tumor models	Enhancing in vitro DC maturation, antigen cross-presentation, T cell cross-priming Enhancing in vivo lymph node delivery, uptake, and DCs maturation	Animals showed a decent therapeutic response upon the first cycle of immunization with the nanovaccine and underwent a second cycle together with anti-PD-1 therapy Suppression of tumor relapse	[114]
**NTV** (p[OEGMA4-DMAEMA22]-p[MA] 30 with conjugation of NDP or PDP with an acid-sensitive acetal bond, OVA241–27)	In vitro/Animal model Human papillomavirus-E6/E7 and B16F10-OVA and tumor mice models	Altering the vaccine morphology from nanospheres into nanosheets in the endosomal environment with high acidity Disrupting the endosomal membrane’s antigen delivery into the cytoplasm Directing anti-tumor immune responses following re-assembly of the nanosheets by specific inflammation pathways activation	Inducing a robust anti-tumor immune response without significant systemic toxicity Combining the anti-PD-L1 antibody and NTV could prolong the survival time of animals Tumor regression in almost half of the studied mice	[115]
**Neoantigen-loaded** **Nanovaccine** (Acid-activatable polymeric conjugate of the DMXAA and neoantigen)	In vitro/Animal model B16-OVA melanoma and 4T1 breast tumor	Activating the STING pathway Accumulation of nanovaccines at the lymph nodes Inducing neoantigen uptake by DCs	Enhancing cancer immunotherapy Stimulating IFN-β secretion Boosting of neoantigen-specific T-cell priming Improving anti-tumor responses following combination therapy with anti-PD-L1 antibody and the nanovaccines in a 4T1 breast cancer model	[116]
**SeaMac** (Polymer NPs, neoantigen)	In vitro/Animal model colon carcinoma 26 (CT26) and B16-F10 tumor models	Promoting DCs’ maturation DCs accumulation in lymph nodes Expanding cytotoxic CD8+ T cells	Inducing a robust anti-tumor immune response Abscopal effects in CT26 and B16-F10 tumors Increasing survival time	[117]
**LrTL** (Trichosanthin, legumain, liposome)	In vitro/Animal model Lewis’s lung cancer (LLC), B16-F10, intracranial LLC xenograft, and CT-26 colon cancer	Activating DCs Releasing IFN-γ, TNF-α, and IL-12 Inducing a powerful CD8+ T cell response Elimination of TAMs Inhibiting immunosuppressive effects TME remodeling.	Inducing a powerful anti-tumor immune response in vitro and in vivo	[118]
**hEX@BP** (Black phosphorus quantum dots and exosomes)	In vitro/Animal model LLC cells	Stimulating differentiation and maturation of DCs Infiltrating effector T cells	A combination of the hEX@BP and photothermal therapy had long-term photothermal therapy performance along with tumor temperature elevation in vivo	[119]
**MSNs-ABC@PDA-OVA** (Mesoporous silica NPs, OVA, photothermal agent polydopamine, and antigen release promoter ammonium bicarbonate)	In vitro/Animal model Melanoma	Recognition of released antigen following laser irradiation Enhancing DCs maturation and activation Facilitating migration of the activated DCs to tumor-draining lymph nodes	Stimulating strong anti-tumor immune responses A single dose of MSNs-ABC@PDA-OVA in combination with a single round of photothermal therapy effectively eliminated melanoma tumor cells Increasing IFN-γ and TNF-α Creating a robust immunological memory Preventing tumor recurrence and lung metastasis	[120]
**PEI-functionalized GO transformable hydrogel** (Polyethylenimine, graphene oxide and R848-laden)	In vitro/Animal model B16-OVA cells	Generation of OVA mRNA Protecting the mRNA degradation Facilitating targeted delivery to lymph nodes Increasing the number of specific CD8+ T cells Generating antigen-specific antibodies	Inhibiting the tumor growth and metastasis	[121]

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
