# Peer review of "Nanovaccines for Cancer Prevention and Immunotherapy: An Update Review"

_cancers, 2022, doi:10.3390/cancers14163842_

Round 1

Reviewer 1 Report

Major comments: 1. A glossary/box with a list of all the abbreviations in the review. 2. The authors should discuss examples of significant clinical trial/trials or immune responses for each approach, in each section of the manuscript. 3. The paragraphs after table 1. do not transition smoothly. The authors should mention why they are discussing those cases in detail.  4. In the examples discussed in section 5, the authors should write in detail with statistics, such as how much of an improvement in immune infiltration, survival, etc. to make a strong case for Nanovaccines. 5. Multiple grammatical errors throughout the manuscript needs to be corrected.   Minor comments 1. Delete the word elicit in line 78 or rephrase. 2. In the section Antigens, describe the difference between TSA and TAA at the beginning. 3. The authors should cite references for claims such as the ones made in line 261-262.   4. The authors should specify the CD11b+ cell type in line 294.

Author Response

Reviewer 1#

Major comments:

  1. A glossary/box with a list of all the abbreviations in the review.

# Author’s response: We appreciate your feedback and suggestions. A list of all the abbreviations has been added to the manuscript.

2.The authors should discuss examples of significant clinical trial/trials or immune responses for each approach, in each section of the manuscript.

# Author’s response: According to your comment, some data have been added to the most important sections.

  1. The paragraphs after table 1. do not transition smoothly. The authors should mention why they are discussing those cases in detail.

# Author’s response: Thanks for your suggestion. An introductory paragraph mentioning the importance of using nanosystems in cancer vaccines has been added to this section.

  1. In the examples discussed in section 5, the authors should write in detail with statistics, such as how much of an improvement in immune infiltration, survival, etc. to make a strong case for Nanovaccines.

# Author’s response: Thanks for your suggestion. Section 5 and Table 1 have been revised according to the comment.

  1. Multiple grammatical errors throughout the manuscript needs to be corrected.

# Author’s response: The manuscript's text has been revised based on your comment.

Minor comments:

  1. Delete the word elicit in line 78 or rephrase.

# Author’s response: It has been corrected.

  1. In the section Antigens, describe the at the beginning.

# Author’s response: the difference between TSA and TAA has been described initially.

  1. The authors should cite references for claims such as the ones made in line 261-262.

# Author’s response: It has been corrected.

  1. The authors should specify the CD11b+ cell type in line 294.

# Author’s response: It has been corrected.

Reviewer 2 Report

Fang X and et al. presented the review titled with Nanovaccines for Cancer Prevention and Immunotherapy: An Update Review. This review is clinically interesting and suitable for publication in this journal with the following comments:

1.      Antigen and adjuvants are not nanovaccines but the concents. Therefore, the subtitle on Line 77 should be“What Are Nanovaccines Composed of?

2.      What about TTAs on Line 117-118 and 531?

3.      Figure 1 should be present fully in Section 2. Thereby, Section 4 would be simply reviewed in this figure.

4.      The authors should identify the study, preclinical, I-III clinical trial and clinical treatment in Section 5.

Author Response

Reviewer 2#

Fang X and et al. presented the review titled with Nanovaccines for Cancer Prevention and Immunotherapy: An Update Review. This review is clinically interesting and suitable for publication in this journal with the following comments:

  1. Antigen and adjuvants are not nanovaccines but the concents. Therefore, the subtitle on Line 77 should be “What Are Nanovaccines Composed of?”

# Author’s response: Thanks for your time and valuable comments. It has been corrected.

  1. What about TTAs on Line 117-118 and 531?

# Author’s response: It means tumor-associated antigens (TAAs). The mentioned items were typographical errors and have been corrected.

  1. Figure 1 should be present fully in Section 2. Thereby, Section 4 would be simply reviewed in this figure.

# Author’s response: Thanks for your comment. Figure 1 has also been cited in Section 4.

  1. The authors should identify the study, preclinical, I-III clinical trial and clinical treatment in Section5

# Author’s response: The updated details of the reviewed studies in Section 5 have been identified in Table1.

Reviewer 3 Report

In this interesting review, Fang and colleagues focus their attention on the impact of nanovaccine in immunotherapy. The manuscript is well written and the figures are really evocative. A point that could be improved is related to the introductive part: in particular, paragraph number 3 is too poor to give an exhaustive overview of the different kinds of nanocarriers. It would be useful to address this point and improve the details in order of the nature, the current use, and the potential therapeutic applications. The field of research focused on exosomes, for example, is in continuous evolution, and even if the related section is well written, the references could be updated with more recent works related to their prognostic value in terms of disease staging (i.e. PMID: 34839044) and the needs for new technologies for the association of a specific marker with an exosome subtype and the exosome subtype to a particular function and/or group of functions ( PMID: 35141731 and others). The same approach should be used to revise the other "objects" in paragraph 3. 

The overall consideration goes in the direction of a really good paper and I feel that also taking into consideration the above comments the paper could be appreciated by the Cancers readers. Good luck.

Author Response

Reviewer 3#

In this interesting review, Fang and colleagues focus their attention on the impact of nanovaccine in immunotherapy. The manuscript is well written and the figures are really evocative.

A point that could be improved is related to the introductive part: in particular, paragraph number 3 is too poor to give an exhaustive overview of the different kinds of nanocarriers. It would be useful to address this point and improve the details in order of the nature, the current use, and the potential therapeutic applications.

# Author’s response: I would like to thank you for your time and comments. Section 3 has been improved based on your comment.

The field of research focused on exosomes, for example, is in continuous evolution, and even if the related section is well written, the references could be updated with more recent works related to their prognostic value in terms of disease staging (i.e. PMID: 34839044) and the needs for new technologies for the association of a specific marker with an exosome subtype and the exosome subtype to a particular function and/or group of functions (PMID: 35141731 and others). The same approach should be used to revise the other "objects" in paragraph 3. The overall consideration goes in the direction of a really good paper and I feel that also taking into consideration the above comments the paper could be appreciated by the Cancers readers. Good luck.

# Author’s response: The mentioned studies have been added to Section 3 to improve this section.

Round 2

Reviewer 2 Report

All is okay.